# Urban Land Monetization-Driven Land Use Orientations: An Insight from Land Lease Prices in Addis Ababa

Amanuel Weldegebriel [1,2,*] , Engdawork Assefa [2], Meron Tekalign [3] and Anton Van Rompaey [1]

1. Department of Earth and Environmental Sciences, University of Leuven, Celestijnenlaan 200E-2411, 3001 Leuven, Belgium; anton.vanrompaey@kuleuven.be
2. Center for Environment and Development, College of Development Studies, Addis Ababa University, Addis Ababa P.O. Box 1176, Ethiopia; engdawork.assefa@aau.edu.et
3. Center for Environmental Sciences, College of Natural and Computational Sciences, Addis Ababa University, Addis Ababa P.O. Box 1176, Ethiopia; meron.tekalign@aau.edu.et
* Correspondence: amanueltadesse.weldegebriel@kuleuven.be

**Abstract:** Urban land leasing is a land monetization strategy that was introduced in 1991 by the contemporary regime. Since then, urban center slum demolitions and their replacement by high-end commercial buildings and urban peripheral low-cost residential condominium expansions have been common occurrences in Addis Ababa. Land rentiers quote extreme land prices at the city center and relatively low prices towards the periphery. Therefore, it has been hypothesized that urban land supply and land prices are determinant factors for urban land use orientations, which have pushed low-end groups towards the periphery. Therefore, based on a lens of land rent theory, 1524 land lease prices and 1038 randomly selected land parcels using Google Earth were used to evaluate locational trends in land prices and land use orientations, respectively. This study revealed that there are significant variabilities between government benchmark land prices and actual quoted land prices. Because of the high rent gaps at the city center, significant land price quotations were recorded, and this overlaps with the urban center slum demolitions and slum resident resettlements at low-cost residential condominiums in the urban periphery. In the first 5 km from the urban economic center, land prices show a declining trend towards the periphery. The central business district is dominated by slums partially under demolition and high-end commercial buildings, while the periphery is dominated by high-rise low-cost residential condominiums. Therefore, the distance from the city center was found to be an explanatory factor of urban land prices. The contributions of other urban utilities to land prices, such as access to transportation routes, could be a future research area.

**Keywords:** urbanization; land rent; urban land monetization; land lease; urban land use; Ethiopia; Addis Ababa

## 1. Introduction

Land rent theory is a geographical economic theory that demonstrates how urban land price and land use varies across a distance expanse from the central business district (CBD) to the outermost area of the city. It also takes into account that different rentiers are competing for land that is closer to the city. The land supply (space production) rate by the government determines the nature of urban land rent [1]. Land is responsive to pressures of supply (production of space) and demand (rentier preferences and rent potential) [2]. If the government produces less space, the land rent rises due to land scarcity. The rent theory explains the amount of money paid for land use, excluding placed capital such as buildups and structures. Land rent is attributed to not only landowners' land price offers and the competition between land use actors, but also the land where production takes place [3].

According to this concept, the theory of land rent explores rent pay for land use only, so rent value is dynamic across a land use gradient and requires land use analysis. Therefore, differential land rent could occur based on the use value of a land parcel at a certain location.

The rent gap is the range of value generated from a given parcel and the potential value that could be earned with capital investment and redevelopment [4]. Thus, the rent gap has an influence on urban land use changes such as urban renewal, demolition, and the sociospatial structure [5]. One scholar explored personal preferences for a given location, but this factor is not as powerful as the capital and property development needs [6]. The disparity between these theoretical perspectives was empirically tested [7] to prove Smith's rent gap theory, which advocates that personal preferences are not as influential as capital in the urban change process. Evidence has shown that rent gaps exist both in declining neighborhoods as well as in consistently poor areas, when there is a substantial capital investment in the urban fringe. This is also in line with the uneven development theory [8], where a more developed part of a city can cause changes in a less developed part of the city. The rent gap hypothesis defines the decline of the inner part of the city, which results from a gap between what the landowner is earning from a land parcel and what the owner could have accomplished if the land were put to its best use [4]. On the other hand, the best use of land can also be determined based on its location [9].

Inner city areas that are either vacant or occupied with low-income households are likely to be transformed into residential or commercial areas for high-income groups [10]. Inner urban slum demolition and residents' peripheral relocation in Sub-Saharan African cities, targeted by urban developers for economic and political goals, are attributed to the loss of use value for low-income classes [11]. Under the market-based neoliberal urban development approaches, the urban restructuring process causes displacement and spatial divisions [12].

Among urban governance constituents, land monetization is one of the key strategies that can be pursued by urban municipalities [13] to boost accumulation, but this can dispossess residents. Urban land commodification is also associated with residents' displacement from their domicile, when the land is treated as a purely financial asset [14,15]. As urban development strategies are supposed to be empirical evidence-driven, an analysis of uneven development and rent gap needs to be carried out across a land use expanse ranging from the urban centers to the urban fringes. This in turn will be useful for urban planners to guide a city's growth through location-based appropriation. The government, a powerful land rentier and a provider of land use strategies (such as allocating lands for varied land uses, transport policy, and road network systems) and other land rentiers' preferences and their land price quotations shape urban growth and dynamics [16]. This also ultimately affects the livelihood of residents and their access to urban amenities [17]. Decreased urban renewal due to a high rent gap can be referred to as gentrification, which can cause displacement and thus a deterioration of life quality [11]. Gentrification is attributed to the growth of a socioeconomic class that replaces low-income classes [18] and entails land use changes. The rent gap theory, which identifies land rent as actual and potential appropriations made from the land, has also connected rent gaps with gentrification [4]. In a gentrification theory, as the city expands outward, residents incur high transportation fares, and the costs of building new houses increase, so buying old houses at the city center and renovation are economically low-priced options [4]. Therefore, urban dynamics are interlinked with market-oriented land regulations and with restructuring the respective governance and associated urban development programs [19].

Rent is a surplus (profit) that is borne out of certain advantages of specific land economic activity. This depends on the ability and willingness of the rentier to pay [20]. Rents are highest at the city center, where retail and commercial activities are carried out to generate more income. Rentiers may not have the same capacity (land use) to rent specific locations at certain distances from the city center. Lands that are the most accessible and that generate the highest income receive the highest rent, which declines along with the distance from the center as a result of the marginal cost of distance for each activity [1]. Thus, distance from the CBD determines the bidding rent of different rentiers for different land uses. The function derived from both the distance from the city center and the respective

bidding rent that rentiers are willing to pay to achieve a given level of utility is referred to as the bid rent function.

Land rent theory assumes a central business district that represents the most desirable location with a high level of accessibility. For example, the surrounding areas within a radius of 1 km have a surface (S) of about 3.14 square km (S = $\pi D^2$) [1]. Under such circumstances, the rent is a function of the availability of land, which can simply be expressed as 1/S. At zero distance, the rent is the highest; thus, as we move away from the center, the rent drops substantially since the amount of available land increases exponentially. Therefore, when there is more land available on which to bid, the price usually decreases. This rent/distance relationship has an effect on land use [1]. The commercial land use is the steepest rent curve, which implies that a sharp reduction in rent occurs as the distance from the CBD increases, and commercial land use is closest to the center (see Figure 1).

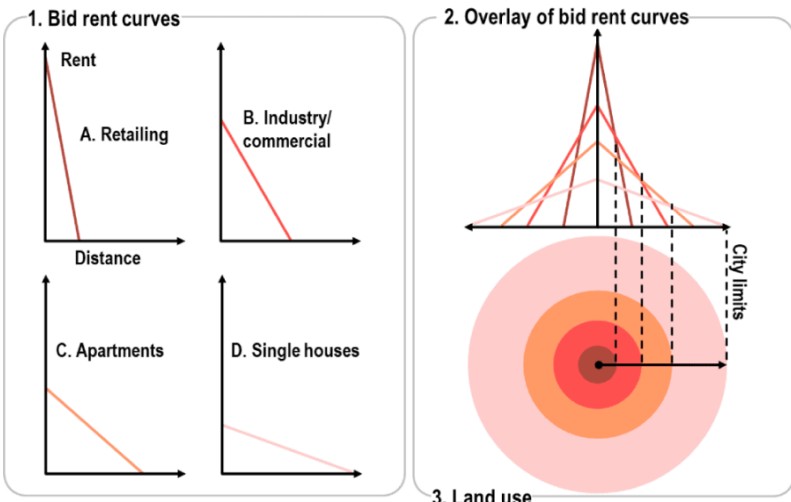

**Figure 1.** Land rent and land use: overlapping bid rent curves of all urban economic activities for a concentric land use pattern, consisting of retail in the CBD, industry/commercial areas in the next ring, apartments in the next, and then single houses in the next, according to Reprinted with permission from ref. [1], Copyright 2013 Routledge.

The land use model application varies depending on the age, size, and locational setting of the city. Concentric cities are smaller and older, while polycentric cities are larger and trending in urban development [21,22]. The urban land rent and land use models that are applied in the developed world may represent cities in the developing world such as Sub-Saharan Africa, where an informal landscape of shantytowns represents a land use structure that is not effectively captured by conventional land use models [1].

In the rapidly expanding cities of Sub-Saharan countries, as rent increases at the most accessible places, remote locations are being used by people with low economic status [23,24]. The CBD, where a small amount of land is usually available, is occupied by high rise buildings for space use maximization. Therefore, increased accessibility to central urban functions generates more rent and can cause gentrification [25]. On the other hand, larger and low-density houses could be expanded in the suburban areas due to low land rents. This may be useful to explain the city's inner reorganization and outward expansion. The complex city nature could be captured through a concentric circle for rent conceptualization based on Burgess' model, who divided a complex city into concentric circles to describe land uses from the center to the suburbs [1,21]. Similar to distance-induced land prices, access to socioeconomic amenities decreases towards the urban periphery [26], and less powerful rentiers settle in locations that are commensurate with their bid rent capacity [27]. The space production (number and size of land parcels supplied and their respective locations) is also an important factor in geographical theory [21]. Some scholars have indicated that regulatory frameworks associated with land monetization shape city dynamics [28,29]. This hypothesis asserts that land rentiers within existing government

regulations and land monetization strategies can determine the structure of a city's urban land use dynamics, which also dictates who is settling where.

Therefore, we argue that land monetization strategies have contributed significantly to the extent of urban core slum demolition, outward low-cost residential expansion, and other associated patterns of urban change. This study aims to examine how the land monetization strategy of the city has affected urban land use changes by comparing the urban dynamics (slum clearance at the center and low-cost housing expansion at the periphery). This is evaluated within the lens of land rent theory, which postulates that urban land price is negatively correlated with the distance from the city's designated economic center [21]. Therefore, urban land use changes before and after land monetization were analyzed to explain urban changes.

The characteristics of urban land prices at contrasting locations from the city center to the periphery were also analyzed to determine if land price patterns match the urban changes (buildup demolitions and expansions at contrasting locations). Additionally, the urban space supply and its effect on quoted land lease prices were examined at contrasting locations. Moreover, the comparability of government-determined benchmarking prices with actual rentier-quoted prices and their patterns within the concept of a concentric city were evaluated. Given that the urban management's land price benchmarking is inconsistent with the actual prices offered by bidders, analyzing this situation might fill the information gap in urban planning and help land management make evidence-based decisions for realistic benchmark land pricing.

## 2. Materials and Methods

### 2.1. Study Area

Addis Ababa was identified as the capital of Ethiopia by the Emperor Menilik the II and the Empress Taitu in 1886 [30,31]. It is located in East Africa (between $8°53'46.92''$ N latitude and $38°55'52.22''$ E longitude). The city has an area of 540 km$^2$, of which 18.174 m$^2$ is rural, and its altitude ranges from 2000 to 2800 m.a.s.l [32]. The city is at the foot of the Entoto range (altitude 2900 m), lowering to 2300 m in the southern periphery toward the Akaki Plains [33]. The city is divided into 10 subcities and 116 Weredas (districts).

In Addis Ababa, since its foundation in the late 19th century, several master plans have been issued with successive revisions, in most cases with the support of foreign planners and architects [31]. The city is one of the late-urbanizing Eastern African cities [34], where urbanization has been prominent over the past two decades. The same period coincides with the shift of national policies from rural-centered, agriculture-development-led industrialization (ADLI) [35] to urban-based development, where an IHDP (Integrated Housing Development Program) was devised to tackle unemployment and increase access to affordable housings [36]. Urbanization and housing development programs were embedded within the SDPRP (Sustainable Development and Poverty Reduction Program) (2002–2005) and the Plan for Accelerated and Sustained Development Program (PASDEP) (2005–2010), and the IHDP was part of this. In the last decade, the Growth and Transformation Plan (GTP) I (2010–2015), GTP II (2015–2020), and the Addis Ababa Structural Plan (2017–2027) were developed [36].

Addis Ababa has become a challenge in terms of affordable residential housing, and urban land became a capital accumulation through speculative land holding and transactions, and property development. According to Ethiopian law, land is state-owned, and citizens have the right to use-right transfers according to the land leasing proclamation. The inner core of Addis Ababa is dominated by slum neighborhoods, where housing is also state-owned and rented to residents. The residents do not have the right to amend housing structures or transfer land to a third party. Only the government can take advantage of the growing rent gaps at the city center. Therefore, the government in its urban renewal program demolishes the urban center slums to produce spaces for land-leasing for commercial, residential, or mixed housing types [37].

To investigate the role of the land lease price distribution on the intra-urban changes and land use orientations, the city has been divided into three concentric land zones, as indicated in Figure 2. Therefore, locational distribution patterns of land lease prices were followed to delineate these land zones. Based on the urban land lease price data and the distance from the center, the study area was divided into three land zones. The radius of concentric circles was based on the differentiated topological structure of the city, where the slums are concentrated within the first 5 km, while the transitional zone ranges between the slum quarter and 5 km away from the slum. The suburban zone, where informal construction and low-cost condominium expansion are occurring, is beyond the transitional zone (>10 km away from the city center). This was based on its economic, sociocultural, and political importance. Evidence from satellite images acquired in the early 1970s indicates that the city center designated by the municipality is closer to where the city started. The state palace and other national and international political institutes, such as the United Nations Economic Commission for Africa, the African Union, and various historical buildings, are situated in the vicinity of this center. The radius was measured from the city center, which was identified based on the designation of the municipality. This designated economic city center was also justified in the analysis as economically the most important area, where the highest prices are recorded.

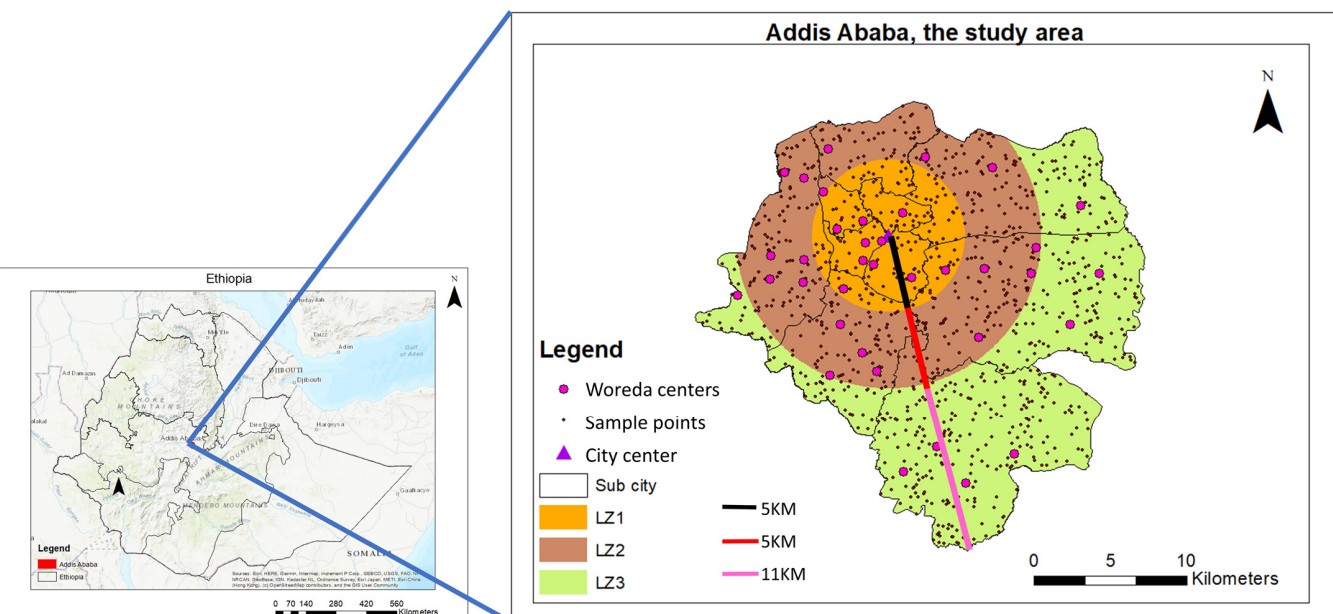

**Figure 2.** The study area (the capital of Ethiopia, Addis Ababa) geographically located within the central highlands of the country. The city is divided into three land zones: at every 5 km for the first two land zones and at the 11th km for the third land zone. The Woreda is the smallest official administration unit and equivalent to a district, and LZ refers to land zone. The city's economic center is based on the municipality's designation.

### 2.2. Land Price Spatial Analysis

Addis Ababa does not have a land price value map. Instead, it has a benchmark land price map based on a land zonation that is inconsistent with the contemporary land prices quoted by the buyers. Therefore, the land lease prices (bid winners' quotations) that were announced between 2013 and 2017 were used to analyze the urban land price locational distributions. These land prices within the time range noted were available at the Integrated Land Information Center (ILIC website). Since 2017, land leasing has been halted by the city administration. Therefore, this land lease data that accounts for 1524 land parcels were obtained from that [38] website. The land lease price comprises the winners' land price quotations, a percentage of the first installment, the land parcel size, the benchmark price,

and the location in the city (both the suburban city and the Woredas/districts). These were the criteria used by the municipality to evaluate bidders' lease price quotations. Land lease prices that had been available in Ethiopian Birr (ETB) currency were converted into USD based on the exchange rate archive from [39].

These land prices were not georeferenced, except their identification was based on the Woreda location. Thus, each Woreda with its land lease price was represented by its centroid, analyzed by converting Woreda features into points. The Woreda centroid land price is the mean of land lease prices quoted by the bid winners in a specific Woreda. This mean land price for each Woreda supports the land price analysis across a distance gradient from the municipality designated city center to the periphery. Each centroid's distance from the city center was also calculated. Therefore, the land prices were compared between all Woreda centroids that are scattered across the city (both at the city center and the periphery). Moreover, the benchmark prices for bidding that are usually determined by the government were also contrasted with the actual land buyer's quoted prices across predetermined distances from the city center to the periphery. For the land price, the trends in the distance from the city center as well as the locational significance in the differences and land price mappings were performed using R-Studio [40] and the ArcMap geographical information system [41], respectively. Based on the city's topological characteristics (slums, transitional areas, and peri-urban land areas), the city was divided into three land zones, as depicted in Figure 1. The next step was to perform a one-way ANOVA for each combination of land zones separately to determine if the land price per m² varies between land zones. One-way ANOVA and F-test statistics were used to analyze the variances between the aforementioned land price sample means for the respective land zones. The one-way ANOVA test preconditions were the test **homogeneity** of land price variances, the test price distribution **normality** of the within-cell residuals, and the influential land price **observations** checked before performing the analysis. For this case, the ANOVA was performed with the Tukey HSD method.

### 2.3. Land Use Spatial Evaluation

To evaluate the land use patterns of the city, 1038 randomly distributed points were each selected with 100 m radius circles. This 100 m selection was determined based on Google Image's visual comfort for easy characterization of the dominant land use. The random urban land parcels that fell within the administrative boundary of Addis Ababa were classified based on their dominant land cover using Google Earth. The land use categories were identified into nine classes (public, open, condominium, regular housing, informal housings, slums, commercial, apartments, and industrial), depicted and explained in Figure 3. The land use distribution is also depicted with the distance from the city center. The limitation with this approach to land use evaluation is the lack of land price information associated with each land use, as the land lease prices compiled from the ILIC were not georeferenced; therefore, the maximum resolution is a Woreda-level identification. Land use evaluations were followed by ground truthing by using 10% (103 land parcels) of the randomly selected land parcel samples. To examine the land use classification accuracy, the kappa index of agreement (KIA) was used. Based on proportional sampling, from a total sample of Lz1 = 154, Lz2 = 410, and Lz3 = 474, random samples of 15, 41, and 47 points were randomly selected from each land zone, respectively. This selection accounts for 10% of the sample points within each land zone. Based on these resampled points, physical evaluations were carried out to determine if land use classifications based on Google Earth images hold true. Therefore, the KIAs for the three land zones were 0.89, 0.96, and 0.97, respectively. Comparatively, the lower accuracy at the LZ1 was due to the topological similarities of old roofed and congested buildups at the inner core. However, the overall evaluation was accepted as per the KIA [42,43].

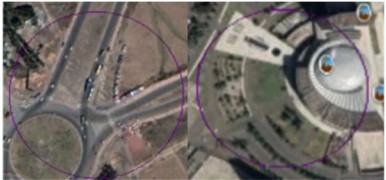
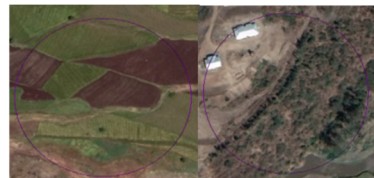
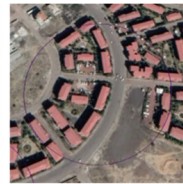
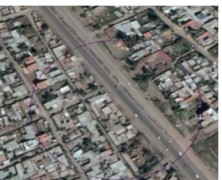

**Public**
Occupied by common use areas and public institutes with national and international entities

**Open**
Non-buildup area covered by agricultural land or shrub

**Condominiums**
4 to 5 story low cost residential condominium housings

**Regular housing**
0-2 story residential housings

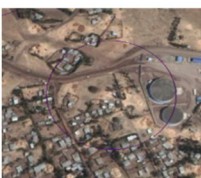
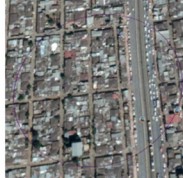
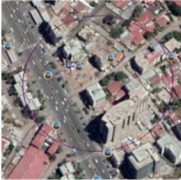
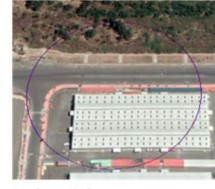
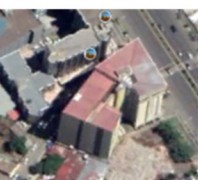

**Informal housings**
Un planned scattered residential buildups, with poor infrastructures

**Slums**
Compacted, and decayed neighborhoods, usually with continuous dark gray roofs

**Commercial**
Multistory buildings for businesses, usually located in the main roads sides

**Industrial**
Low-rise warehouses over extended areas for manufacturing

**Apartments**
Multistory expensive residential buildings for rent

**Figure 3.** The land use classification categories, based on the land parcels falling within the sample point circles. Source: OpenStreetMap features (https://wiki.openstreetmap.org/wiki/Map_features, accessed on 14 July 2021) and Google Earth features (https://earth.google.com/web, accessed on 18 July 2021) were used to define the identified land use features to conduct classification, which was also verified based on in situ observations to minimize error.

### 2.4. Urban Slum Demolition and New Expansions

Housing demolitions and urban expansions in the city were assessed using high-resolution satellite imagery (USGS, 2020) coupled with practical field observations for validation. This was carried out with a benchmarking of urban land (buildup and non-buildup) changes before and after urban land monetization strategies. The demolition sites were explored and delineated using Google Earth, which was triangulated with government urban renewal program data. This exploration covered satellite imagery evaluations by switching between the year when the urban land lease was enacted (1993) and 2020. Based on the integrated household development, the inner urban slum was targeted for demolition and renewal.

To detect the extent of urban expansion after urban land lease introduction, freely available Landsat 8 images were used and processed to enhance their quality in terms of resolution (30 m). The 1993 images were composited for RGB (4, 3, 2), and the 2020 images were composited for RGB (5, 4, 3). Further analysis was done to classify land use via a supervised classification technique with sufficient training samples. The supervised binary land use classification for the buildup and non-buildup was performed with ArcMap GIS 10.7 [41]. Figure 4 shows the general procedures.

| Images acquisition | Image processing | Change detection |
|---|---|---|
| Landsat | radiometric enhancement, stacking, supervised classification | comparison |

**Figure 4.** Land use classification protocol using Landsat images.

The urban expansion after the introduction of these land leasing mechanisms was represented by the buildup expansions between 1993 and 2020. To assess the accuracy of the supervised binary classification of non-buildup (NBU) and buildup (BU), ground truthing was conducted at 200 randomly selected sample points, which yielded a 93.4% prediction accuracy, which is acceptable based on the KIA [42,43].

## 3. Results and Discussions

### 3.1. Spatial Distribution of Urban Land Lease Prices

The mean price value of the aggregated land prices captured at the Woreda level is represented by the Woreda centroids (Figure 5). According to this map, higher land price quotations by the land lease bidders concentrate towards the city center. The economic center of the city is closer to the northwestern part of the capital. Only 36 Woredas out of 116 were subjected to the land leasing program between 2013 and 2017. Unlike the city economic center that is determined by the relative intensity of social, economic, political, and cultural activities in the city, the Woreda centers were analyzed based on the Woreda administrative boundaries geographic centers.

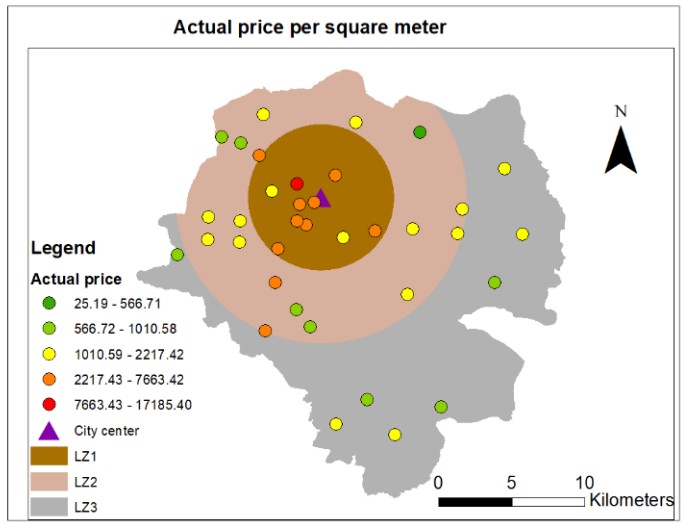

**Figure 5.** Urban actual land lease price (in USD) distribution by Woreda (the smallest administrative unit) centers. Each land price point is the mean value of land prices quoted in the Woreda.

The actual quoted land prices per square meter are extreme with a range of USD 25–17,185. As indicated in Figure 6, extreme price quotations were observed at the most central part of the city. The city center overlaps with four subcity administrations. These subcities are where the capital historically started as a political, social, and economic center. The national palaces, historical sites, churches, and large open markets such as Merkato are located at the city center. The land price at the center is extremely higher compared to other parts of the city, which is attributed to high competition for commercial buildup development over other land uses.

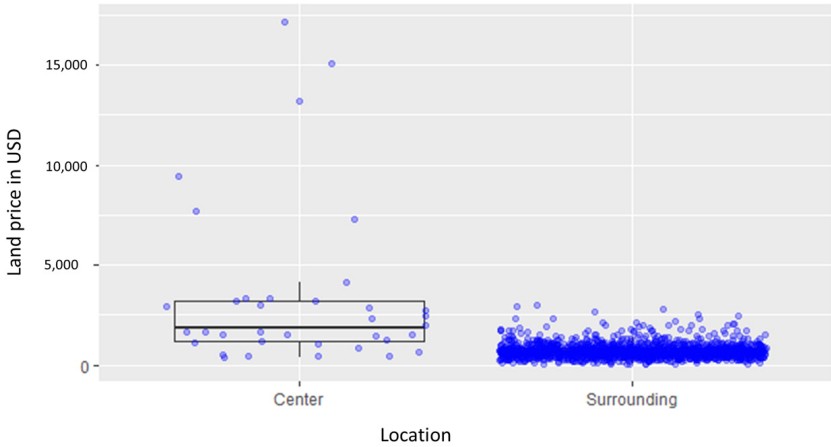

**Figure 6.** Actual urban land lease prices per m$^2$ in USD by categorized locations (the center and surrounding). The center is within the CBD, while the surrounding is beyond the 5 km radius.

The land price shows a declining trend against the distance from the center towards the periphery. The land price value against distance declines in the first 5 km from the center and is characterized by a consistent value beyond the indicated distance (Figure 7). There is also a slight significant price difference between the land price values at distances of 5–10 and 10–16.97 km. There was no land price value recorded beyond 16.97 km. Therefore, the price data analysis based on the aforementioned distances, 0–5, 5.1–10, and 10.1–16.97 km, categorized as Land Zones 1 (LZ1), 2 (LZ2), and 3 (LZ3), respectively, shows significant variability in prices (see Table 1).

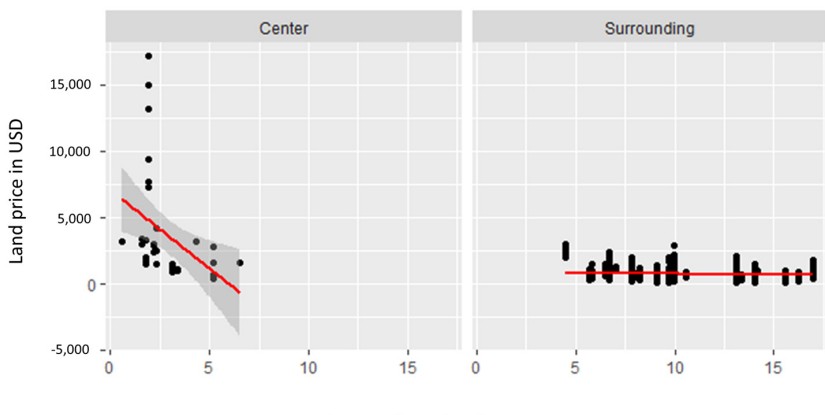

**Figure 7.** Actual urban land prices per m² in USD against distance in kilometers from the economic center and categorized as center or surrounding. The center is within the CBD, while the surrounding is beyond the 5 km radius.

**Table 1.** Descriptive summary of land price per m² in USD for the designated land zones.

| Land Zone | *n* | Mean | Sd | Median | Min | Max |
|:---:|:---:|:---:|:---:|:---:|:---:|:---:|
| LZ1 | 36 | 3829.6282 | 3940.3584 | 2505.2934 | 870.14531 | 17,185.37 |
| LZ2 | 733 | 700.4355 | 405.5964 | 601.8505 | 26.34607 | 2907.736 |
| LZ3 | 755 | 690.5385 | 254.1457 | 664.936 | 25.18587 | 2085.932 |

The Levene test (homogeneity of variances test) returned a significant variation between variances of observations: Df = 2, F value = 182.13, and Pr(>F) = $2.2 \times 10^{-16}$. LZ is for land zone.

Less responsiveness of land price to distance in the last two zones (LZ2 and LZ3) could be due to the existence of other physical and socioeconomic attributes that determine land prices. Extreme land price values were observed only in LZ1 (0–5 km) (see Figure 8), where the commercial buildups are emerging, political organizations are situated, and international companies and real estate are concentrated.

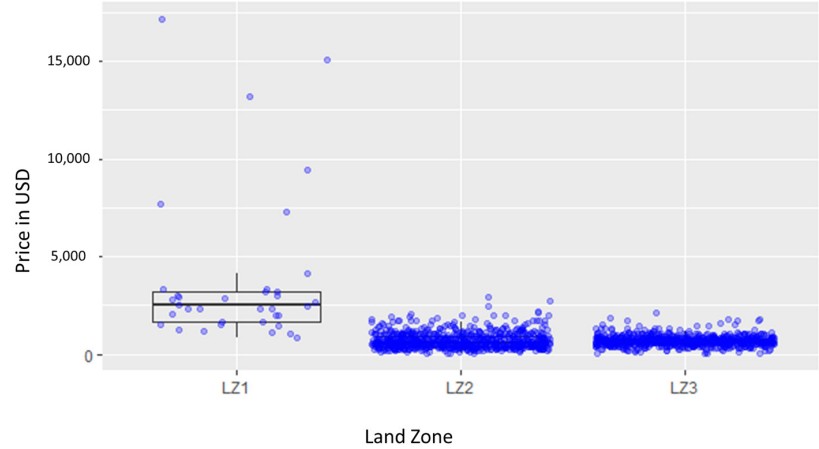

**Figure 8.** Actual urban land prices per m² in USD by land zone.

Therefore, the land price, as is depicted according to the location (center to periphery), showed a declining trend (0–5 km) and was comparatively less responsive as it moved from LZ2 and LZ3, which span from 5 to 16.97 km (Figure 9). In the respective land zones, other land price attributes could be more explanatory than the distance from the center. However, an analysis of variance indicated that there is a significant land price difference between the delineated land zones (Table 1). In this case, price is assumed to capture the land value, which includes several urban land-actor-preferred variables, but such a nonlinear land lease price trend beyond the CBD could be a good starting point for future research on the contributions of distances from basic urban utilities, such as access to transportation routes, to land price determinations.

**Figure 9.** Actual urban land prices per m$^2$ in USD against distance from the center in kilometers and categorized by land zone.

Here, the hypothesis established was that mean land price equality never holds across all land zones ($\bar{x}$ LZ1 = $\bar{x}$ LZ2 = $\bar{x}$ LZ3), where $\bar{x}$ is the sample mean, and LZ is the land zone. The mean land price for the first land zone seems considerably higher than the remaining two land zones, which are comparable to each other (see Table 1).

Given that there was heterogeneity, a one-way ANOVA of land price per m$^2$ means that variability significance did not assume equal variances. Thus, a one-way test that applies when variances are not equal was performed. The one-way ANOVA showed that when unequal variances were returned, there was a significant land price difference between land zones (see significance tests below for each group). Therefore, according to the one-way analysis of means (not assuming equal variances), based on the data of the price per m$^2$ and the land zones, F = 11.476, num df = 2.000, denom df = 90.763, and *p*-value = $3.604 \times 10^{-5}$. This indicates that there is significant land price difference between the land zones.

The next step was checking the normality of the land prices per m$^2$ with the Shapiro–Wilk normality test, which returned a rejected normality (*p*-value < $2.2 \times 10^{-16}$) for the different categories of land zones. To maintain normality, the land price data were logarithmically transformed. To trace extreme land price observations, the Cooks distance versus the observation number was plotted to identify and omit influential land price values. This minimized the effect of few but extreme observations over the majority that fell under a normal distribution. Finally, Tukey's HSD (honestly significant difference) test, a single-step multiple comparison procedure and statistical test, was applied. It was used to find means that were significantly different from each other (Table 2 and Figure 10).

**Table 2.** Significance tests for land price per m$^2$ in USD for the designated land zones.

| Land Zone | diff | lwr | upr | *p* adj |
|---|---|---|---|---|
| LZ2-LZ1 | −1.54201912 | −1.753800107 | −1.3302381 | <2 × 10$^{-16}$ |
| LZ3-LZ1 | −1.46983763 | −1.681474121 | −1.2582011 | <2 × 10$^{-16}$ |
| LZ3-LZ2 | 0.07218149 | 0.007853048 | 0.1365099 | 0.0232759 |

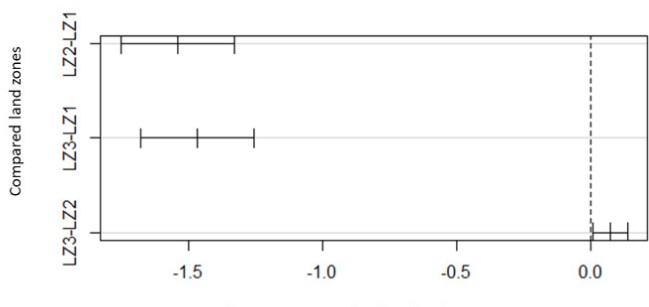

**Figure 10.** Visual representation of mean land price differences between different paired land zones. LZ3-LZ2 is a comparison between Land Zone 3 and Land Zone 2, LZ3-LZ1 is a comparison between Land Zone 3 and Land Zone 1, and LZ2-LZ1 is a comparison between Land Zone 2 and Land Zone 1.

According to the land lease price distributions, declined central slum neighborhoods, due to an increased rent gap, attracted reinvestment in commercial and real estate business development. Therefore, considerable capital flowed into the high-rent-gap [4,44] central neighborhood as a result of foreseen increased potential economic returns, which caused higher prices at the city center. On the other hand, the increased price was attributed to the scarce supply of land coupled with limited land availability at the center. The urban space production affected the supply and demand of land, which ultimately determine land prices. Similar to the highest land price concentration at the center, few but larger plots are comparatively available in the center (see Figure 11).

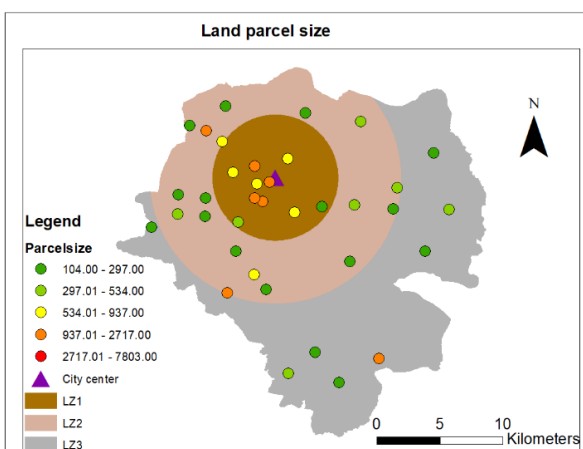

**Figure 11.** Urban land size (in square meters) distribution for land leasing.

Exceptional larger land parcel sizes were also observed towards the southern parts of the city, recognized as an industrial and warehouse zone. Therefore, larger plots better attract commercial or industrial land than residential land. Moreover, out of the total land plots supplied, there seems to be a larger number of plots offered for bidding in the surrounding parts of the city compared to the center. This toughened the bidding competition among the commercial bidders, which resulted in extreme land prices at the center. This is also in line with the production of space-related theories [25] that the

state as an influencing actor in urban land produces urban space that ultimately leads to gentrification, which results in demolition of high-rent-gap locations [4]. Therefore, such a high land rent gap at the center compared to the periphery causes the poor and the middle class to move away from the center, where buildup expansion exceeds the urban amenity distribution [26], which is also a challenge to urban poor residents [45].

High urban land demand coupled with a low allocation of lands for leasing has brought limited access to affordable housing to the urban poor and the middle class [36]. This process leads to accumulation by the government, whose financial accumulation relies highly on land sale [46]. Ethiopia, the second most populous country in Africa with more than 112 million people (2019) and one of the poorest with a per capita income of $850 [47], coupled with rapid urbanization, low access to housing [48], and an unregulated urban land market, suffers from reduced access to urban housing.

On the other hand, a large number of urban land plots were offered for bidding in surrounding districts (expansion areas of the city) (Figure 12). Therefore, the existing open land was subjected to buildup expansions, while slums located at the urban center were demolished for commercial buildup expansion. Therefore, open land has been an easier alternative to allocating urban land parcels for bidding towards the periphery. Such land leasing has been a contributor to the city's expansion since it was enacted by law in 1991.

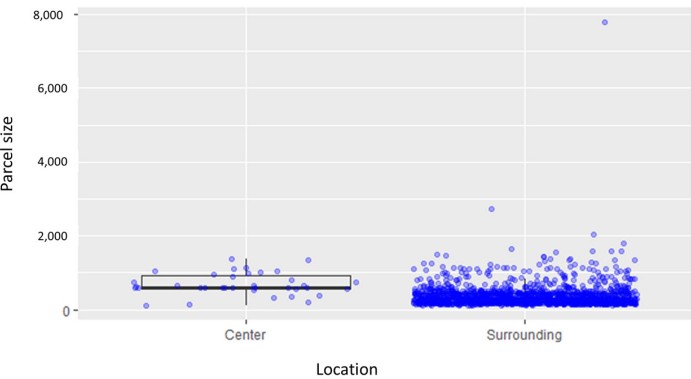

**Figure 12.** Land leasing urban land plot size (in square meters) distribution by their location.

The initial leasing benchmark prices were determined by the municipality's land administration office. Land price determination was based on land zoned into a central district (CBD), a transitional zone (TRZ), a suburban area (SUR), and a green area. In principle, the green land zone is not subjected to land bidding. Land price determination further categorized land zones into different grades, based on preset criteria, where basic infrastructure development was the main factor. The land lease regulation was implemented through land tendering under the provision of proclamation [37]. Within the legal provision of the same proclamation, the city's land administration bureau identified benchmark prices corresponding to each land zone (Table 3). It was identified that the green land zone, which is restricted in buildup, was subjected to the actual land lease system, while no benchmark and actual land prices were identified in the municipality's report.

**Table 3.** Average benchmark price (in USD) per land zone.

| Land Zone According to the Government | Benchmark AverAge Lease Price Determined by the Government | Actual Average Lease Price Analyzed by the Government | Benchmark Average Lease Price as per the Gathered Data (2013–2017) | Actual Average Lease Price as per the Gathered Data (2013–2017) |
|---|---|---|---|---|
| CBD | 63.07 | 469.99 | 13.34 | 6389.02 |
| TRZ | 38.86 | 241.99 | 11.96 | 1854.58 |
| SUR | 12.83 | 179.42 | 38.91 | 733.09 |
| Green | NA | NA | 45.14 | 440.18 |

Source: based on Addis Ababa's municipality, integrated land information center (ILIC) analysis and the actual land lease price data compiled between 2013 and 2017. USD = United States dollars (average annual exchange rate 20.6862 of Ethiopian Birr as of 2015), CBD = central business district, TRZ = transitional zone, and SUR = suburban zone (see Figure 13).

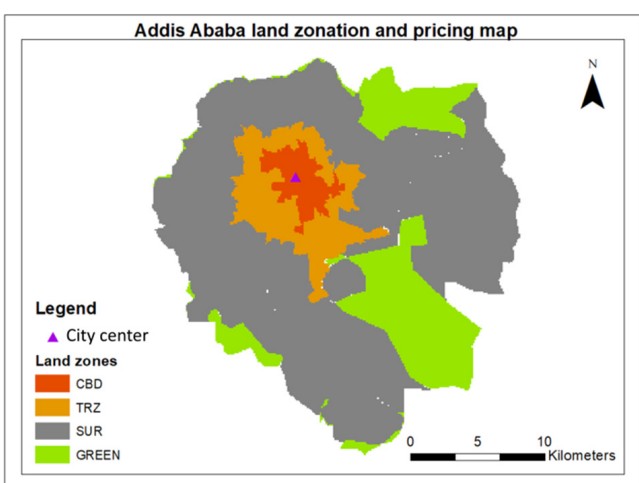

**Figure 13.** Land zonation map of Addis Ababa according to the benchmarking prices. CBD = central busi-ness district, TRZ = transitional zone, SUR = suburban zone, and Green = green area. Source: based on the shape files and benchmark land price data from Addis Ababa's Integrated Land Information Center (ILIC).

Based on the results, due to the existing high demand of urban land in the city, the actual selling price far exceeded the benchmark price set by the municipality [49]. In some cases, the benchmark prices set for the land leases depicted an irregular trend, as there were higher benchmark prices allocated towards the periphery and, on the contrary, lower benchmark prices towards the center. As per the prior analysis, the actual land price quotations negatively correlated with the distance from the economic center, which is in line with the land rent theory. On the contrary, the benchmark land price distribution against the distance from the center shows inconsistent trends with the actual land prices (see Figure 14).

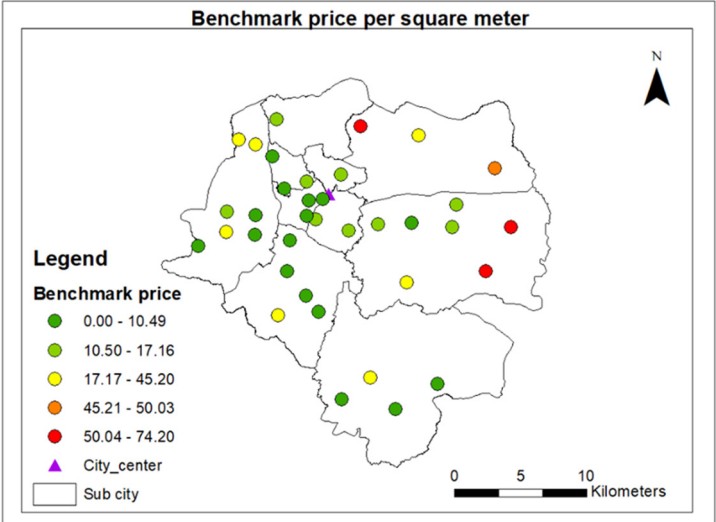

**Figure 14.** Benchmark urban land prices per m$^2$ in USD against the distance from the city economic center by Woredas.

The actual allocated benchmark land lease prices do not show a consistent trend with actual land lease bidders' quoted prices. This is a clear indication that benchmark land prices are not explained by the distance from the center. Even an increasing land price trend was observed towards the periphery of the city (see Figure 15). This land price trend deviation from both the actual quoted prices and the land rent theory shows an existing

gap in benchmark price determination. In addition, benchmarking price errors could be caused in bidding announcements if benchmark prices are not properly referenced.

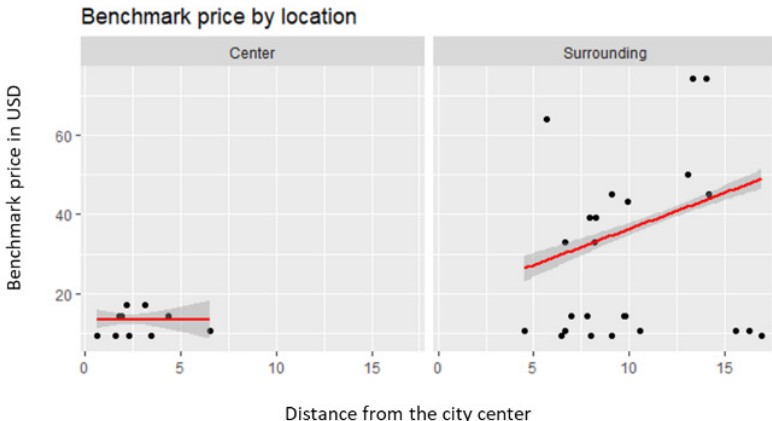

**Figure 15.** Benchmark urban land prices per m² in USD trends against distance from the center in kilometers.

### 3.2. Urban Land Uses

Based on 1038 random land use evaluation points, nine land use classes were identified. Among these land use classes, open land holds the highest proportion, followed by regular housing, i.e., single-story family houses. The smallest land use proportion identified was the apartment class, followed by commercial buildups. Slums, informal housings, and industry seem to have comparable land use coverage in the city. As indicated in Figure 16, the land use distribution across the aforementioned land zones also varies. The slum and commercial land uses are concentrated towards the center (LZ1), which is dominated by slums. In LZ2 and LZ3, the slum class declines to a smaller proportion, while apartment land use does not diminish at all. The land use allocated for public use seems uniformly distributed across all land zones. Open land increases in LZ2 and LZ3. In these land zones, a large number of land parcels were available for bidding (Figure 12). The regular house land use type is more concentrated in LZ1 and LZ2 and declines towards LZ3. More low-cost housing condominiums exist in the surrounding land zones compared to the central zones.

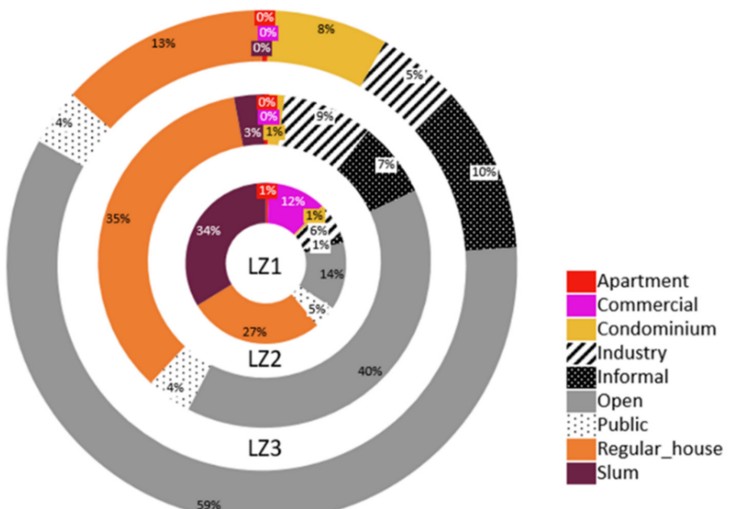

**Figure 16.** Addis Ababa's land use distribution based on the three predetermined land zones.

Given that the land zonation was determined based on the distance from the city center to the land price contrast, it is clear that the land use distribution is affected by this distance (Figure 17). Among the residential land uses, the slums are situated at the

center of the city, while regular housing types and apartments are found in the middle, and informal settlements, though closer to the center than condominium housing, are set up towards the margins of the city. They are built up within available open spaces and on forest-covered hilltops, which are left open in the city's master plan (see Figure 18). In general, condominiums are the farthest residential structures located towards the periphery of the city, where basic urban functions are comparably less developed [26].

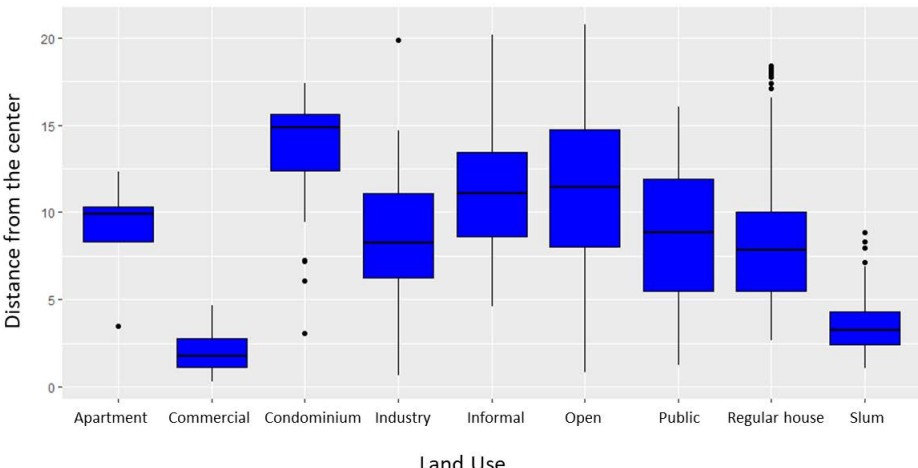

**Figure 17.** The land use distribution plotted against the distance from the city center in kilometers.

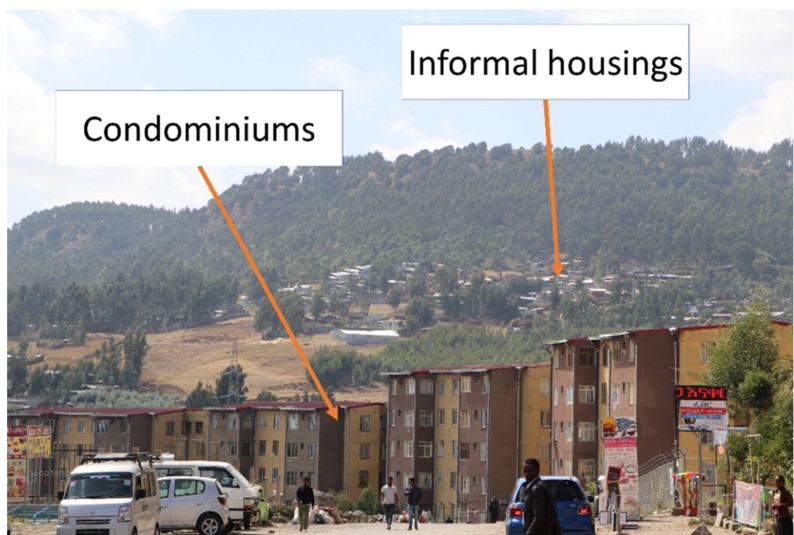

**Figure 18.** Informal housing settlements near condominium expansion areas within the Yeka Abado neigh-borhood. Source: captured by A.T.W. on 13 January 2020.

Nonresidential land uses such as commercial buildups are situated towards the center, while the industrial and warehouse land uses are located in the middle and periphery regions of the city. Public infrastructure seems to be evenly distributed across all land zones, while the coverage of open land uses increases from the center towards the periphery.

### 3.3. Urban Slum Demolition and New Expansions

Delineation of the urban center slum demolition area indicates that the cleared slum entirely falls within the 5 km radius from the urban center. According to Figure 19, which depicts the urban slum clearance and the new expansion, the slum clearance area overlaps with the land zone in which the slums are confined. In the same figure, urban expansion is considerable in the third land zone, followed by the second land zone. The urban expansion measured by the appearance of new buildups between 1991 and 2020 marks the regime

that introduced land leasing. The urban core slum demolition and new urban expansions account for 1.92 and 237.39 km², respectively.

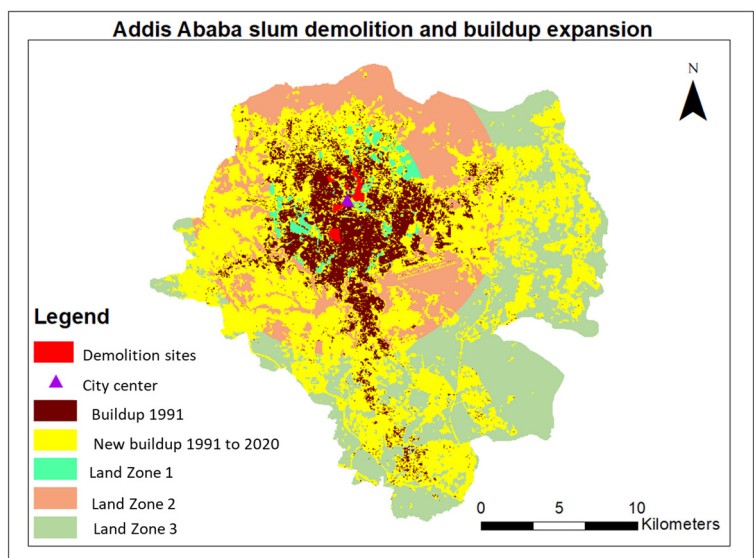

**Figure 19.** Addis Ababa's central core urban renewal (demolition) and urban expansion towards the periph-ery between 1991 and 2020

Figure 20 shows a practical example that evidences the urban core demolitions that were leased for commercial buildup expansions and the peripheral low-cost residential condominium development.

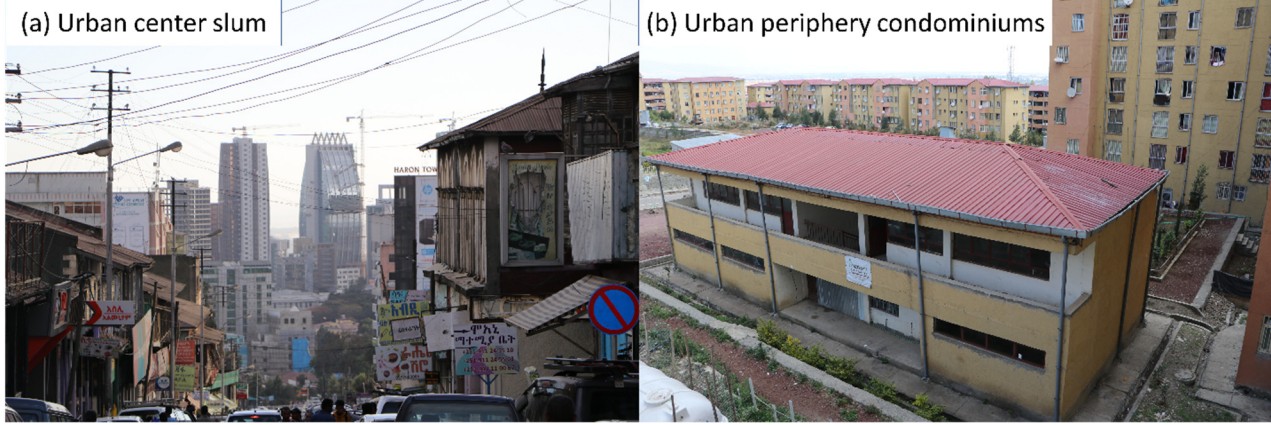

**Figure 20.** The city slum demolition and replacement with the emerging commercial buildups at the center (**a**) and low-cost condominium housing expansions at the urban periphery (**b**). Source: captured by the researcher on 12 January 2020.

Since 1991, the city's administrations have issued policy changes on land and urban development strategies. This juncture has attracted different urban land rentiers to take on their roles in the urbanization process. Land monetization has influenced the urban redevelopment process, and the extent of urban growth has been dependent on different types of actors (developers) that have varying capacity in terms of financial influence, their association with urban political elites, and the fragility of urban governance institutions [46,50]. Urban land is also exceptionally well-supplied to rentiers that are believed to contribute to significant national interests; thus, urban land, by drawing lots for building residences, can be transferred by auctions, negotiations, and allocations [51].

The urban core slum demolition and the following peripheral land leasing are also consistent with the quantitative figures on slum demolition compiled from government

documents [52] and scientific literature [53], tabulated in Tables 4 and 5, respectively. All of the renewal projects are in the central subcities (limited to Arada, Kirkos, and Lideta), and these are clearly visible in Google Earth.

**Table 4.** Urban redevelopment and renewal projects in Addis Ababa.

| Redevelopment Area | Area in Hectares | No. of Housing Units Demolished | | |
|---|---|---|---|---|
| | | Public | Private | Total |
| Sheraton Hotel Expansion (Kirkos Subcity) | 27 | 1358 | 406 | 1764 |
| Sheraton Hotel Expansion (Arada Subcity) | 15 | 998 | 305 | 1303 |
| Meskel Square (Kirkos Subcity) | 3.2 | 172 | 43 | 215 |
| E.C.A. Area (Casanchis, Kirkos Subcity) | 23 | 206 | 74 | 280 |
| Wollo Sefer Area (Kirkos and Bole Subcities) | 9.8 | 324 | 47 | 471 |
| Basha Wolde Chilot (Arada) | 25 | 953 | 342 | 1295 |
| Lideta | 26 | 1134 | 339 | 1473 |
| Total | 129 | 5145 | 1556 | 6801 |

Source: Ministry of Urban Development, Housing and Construction: National Report on Housing and Sustainable Urban Development, 2014.

**Table 5.** Residential housing comparison between the center and the periphery.

| | Central Subcities 2006 | Central Subcities 2016 | Surrounding Subcities 2006 | Surrounding Subcities 2016 | Addis Ababa 2006 | % Total Area | Addis Ababa 2016 | % Total Area |
|---|---|---|---|---|---|---|---|---|
| Condominium (4+) | 13.70 | 122.12 | 185.62 | 2154.46 | 199.32 | 1% | 2276.58 | 4% |
| Single Occupancy | 40.08 | 285.09 | 757.35 | 4651.70 | 797.43 | 2% | 4936.79 | 9% |
| Informal Housing | 1990.29 | 1540.38 | 7646.68 | 6181.14 | 9636.97 | 19% | 7721.51 | 15% |
| Low Rise/Mixed (<4) | 602.81 | 294.23 | 5771.19 | 4892.56 | 6374.00 | 12% | 5186.78 | 10% |
| Total Residential | 2646.89 | 2241.81 | 14,360.84 | 17,879.85 | 17,007.73 | 33% | 20,121.66 | 39% |

Source: Larsen, 2019. Comparison of residential housing types in the central and surrounding areas of Addis Ababa, Ethiopia, in 2006 and 2016. The area covered by buildup types is in square kilometers.

In Sub-Saharan African cities, among other rentiers, the government is the most influential, which is attributed to land nationalization policies. Urban inner core demolition and redevelopment have been achieved by various actors with different levels of power. Commercial builders take part in the regeneration of demolished slums, while the government facilitates slum clearance and space production under its land monetization agenda [54]. This holds true for the Ethiopian government that runs large-scale low-cost condominium constructions at the periphery of the capital. The most underdeveloped slum houses that are located at the city center became the government's target for redevelopment and for land lease transactions based on the different aforementioned conditions. Slum demolition at the center and the farmland-clearing at the expansion area are the main space production strategies for land leasing. Another example in Sub-Saharan Africa could be Lagos city, Nigeria, a densely populated slum quarter where urban redevelopment has been occurring through land monetization but is covered with the city's image transformation agenda [55]. Therefore, states play an important role in producing new spaces for redevelopment, e.g., by promoting basic infrastructure development [11]. Therefore, the results depict that land monetization strategies and land lease price distributions have shaped the city's growth pattern. Such urban responsiveness to the urban governance of land monetization has also been confirmed in other reports [2,19].

The thesis in 2 stresses that urban flagship projects could reshape the dynamics of sociospatial structures from a general perspective. Albeit in an African context, this has also been demonstrated by the low-cost housing expansion megaprojects carried out in the periphery areas of Addis Ababa between 2006 and 2016, a period during which the IHDP was active [53]. The IHDP promoted the urban center slum renewal and low-cost condominium expansions at the periphery for the low and middle economic classes. This resulted in a reduction in informal and low-rise housing at the center and increased residential housing in the surrounding subcities (see Table 5). Along with existing land

leasing, urban land distribution exacerbated the challenge of providing access to affordable housing and led to residential social segregation [56,57]. Therefore, a subsidized and efficient residential housing development strategy against residential segregation could retain some low-income households at the center [58].

The government of Luanda, Angola's capital, had also proceeded with an urban slum renewal program that increased the land price, which excluded the poor from the city and affected the city's land use dynamics in general [59]. In Kigali, Rwanda, an East African city, the government has also been interested in land finance and invested in urban development, which has stimulated private real estate developers and catalyzed urban changes [56,60]. Additionally, the case study on land commodification and its exclusionary power in Ghana has indicated that the ability to pay for a land price that is set by a market determines who owns it and the land use type [61]. This supports the findings of this research, which revealed that land commodification moved high-end groups to the center and displaced the poor to the periphery.

Many factors determine land prices, and these factors include location, proximity, accessibility, land use cost, productivity, land grade, shape, size, and the demand for land, which is derived from the demand for output, e.g., housing [62]. Based on the role of the state in managing urban land supply and prices, land speculation plays a pivotal role in increasing land prices, in both state-controlled and free land markets [62]. The same study indicated that private actors in a land transaction buy land from owners and then sell that land, and these land transferees play a land price inflation role. Land price inflation in turn excludes the poor. Therefore, anti-speculative measures, i.e., real property taxation based on land transfer values, might help city administrations in financing urban infrastructure for a long period of time [63], so as to curb speculative demands and their negative consequences.

## 4. Conclusions

The Ethiopian state land-holding system introduced in 1974 was also maintained by the following regime in 1991. The new regime has shifted its rural-biased development strategy to urban inclusive programs since the early 2000s. The IHDP was coupled with an urban land leasing strategy that shifted the land tenure into a market-based system. The land monetization radically changed the urbanization pattern of Addis Ababa, as the government led low-cost condominium expansions at the periphery and slum clearance at the center to cater to high-end builders for commercial businesses. The space production at the periphery has been substantially higher than in the inner urban core. Such uneven development between the periphery (government social housing and private real estate development) and the center has raised the rent gap at the slum inner core. Such an increased rent gap has exacerbated the demolition of slum neighborhoods at the center.

In the urban core, which fits perfectly with the theoretical land rent curve in the first 5 km (Land Zone 1), land price becomes much higher towards the center, therefore attracting commercial builders. In the peripheral parts of the city, many land plots have been leased, compared to the inner urban core, which has stiffened the competition for land, and this has resulted in extremely high quoted prices at the center. Although there was a significant mean price difference between Land Zone 2 (5.1–10 km) and Land Zone 3 (10.1–16.97 km), land price seems less responsive to distance from the center compared to the inner urban core (Land Zone 1). In addition, the bidders' quoted land prices are much higher than all benchmark land prices predetermined by the government.

Urban land price is a pivotal and decisive factor that determines access to housing for urban residents. In Sub-Saharan cities such as Addis Ababa, the government's financial accumulation relies highly on land sales. Ethiopia, the second most populous country in Africa with more than 112 million people (2019) and one of the poorest with a per capita income of $850 [47], coupled with rapid urbanization, low access to housing [48], and an unregulated urban land market, suffers from reduced access to urban housing. Such high land prices will not be affordable to low-income residents that are concentrated in

the city center slum. This fact is supported with evidence showing that the condominium development megaprojects are situated at the periphery of the city, while the urban core is demolished so that government can benefit from a high land rent. Extremely high land price quotations towards the city center will more likely attract commercial builders, causing the evictions of slum residents and their movement outside the city center, where buildup expansion exceeds the urban amenity distribution.

Based on speculative demand, urban land leasing has been used as a profit-making business by a segment of rentiers. Speculative demand-driven high land prices reverberate into expensive housing. Increasing urban land prices are inevitable unless managed using sustainable urban development tools. Therefore, anti-speculative measures may include increasing capital gains and property taxes based on property value, so as to secure a reasonable price increase and curb speculative demands and their negative consequences. As part of the suggested urban land price stabilization strategy, land lease winners must be subjected to the land development and prohibition of land transfers before development, which is in line with the city's local plan. The prohibition of land transactions without development may also need to be coupled with good monitoring and evaluation systems of land uses (identified land lease prices in the green land zone could be an example of a failed monitoring and evaluation system). If the conventional land monetization strategy is pursued, in addition to worsening the poor's access to affordable housing, it might signify an emerging social segregation by residential location, which is trending in East African cities. Therefore, the municipality's city center needs urban spaces that are intentionally allocated for the low class through affordable schemes instead of a price-driven land transaction system. Higher-density, state-subsidized multistory apartments that accommodate low-income households could be a solution, as it might reduce economic inequality settlement patterns and secure sustainable cities and communities, as per the UN's Sustainable Development Goals.

Addis Ababa is a geographically constrained city that is hard to expand outward beyond its administrative boundary, which is also an exacerbating factor behind the increasing land price. Therefore, the municipality must work collaboratively with the surrounding regional state of Oromia on urban development matters to accommodate the city's large housing demand. Integrating satellite towns in the city's adjacent region could also help to uphold the growing demand for urban land and housing.

**Author Contributions:** Conceptualization, A.W., M.T. and A.V.R.; data curation, A.W. and M.T.; formal analysis, A.W.; methodology, A.W. and A.V.R.; project administration, A.V.R. and E.A.; resources, A.W., E.A., M.T. and A.V.R.; supervision, E.A. and A.V.R.; validation, A.W., A.V.R., M.T. and E.A.; visualization, A.W.; writing—original draft, A.W.; writing—review and editing, A.W., E.A., M.T. and A.V.R. All authors have read and agreed to the published version of the manuscript.

**Funding:** This research was funded by VLIR UOS (https://www.vliruos.be/en/home/1, accessed on 23 March 2022) through the Global Minds Scholarship of KU Leuven University (https://www.kuleuven.be/global/global-development/funding-possibilities/globalminds, accessed on 23 March 2022).

**Institutional Review Board Statement:** Not applicable.

**Informed Consent Statement:** Not applicable.

**Data Availability Statement:** The data presented in this study are available on request from the corresponding authors.

**Acknowledgments:** We are grateful for Global Minds, the VLIR UOS program funded by the Belgian government through cooperation with the University of Leuven. Much appreciation is extended to the Integrated Land Information Center (ILIC) of Addis Ababa, the bureau of land administration. We express our deep gratitude to Addis Ababa University, College of Development Studies (the local partner institution), for their support in the research process.

**Conflicts of Interest:** The authors declare no conflict of interest. The funders had no role in the design of the study; in the collection, analyses, or interpretation of data; in the writing of the manuscript, or in the decision to publish the results.

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
