# Peer review of "Urban Land Monetization-Driven Land Use Orientations: An Insight from Land Lease Prices in Addis Ababa"

_land, doi:10.3390/land11060796_

Round 1

Reviewer 1 Report

The manuscript "Urban land monetization driven land use orientations: An in-2 sight from land lease prices in Addis Ababa" is an interesting paper and with minor revisions should be published. The paper presents evidence of the importance of land monetization phenomenon on urban development. The theories cited are generally known and in this sense the paper does not bring new trends. On the other hand, the procedure of conducting the proof is interesting and valuable. Among the main criticisms I see in the work is the explanation of how the city center is identified. How did the authors define the central zone and how did they reduce it to a focal point? The second main objection concerns the method of defining subsequent radii of circles. Why 5 km? Is it always like that?  Future research should think about using distance buffers instead of circles. Especially since lease prices clearly seem to be correlated in this case with the distance from the city's main transportation routes. Downtown does not seem to be the point. 
 Minor comments relate to the figures and charts. Both Figure 5 and Figure 11 should have the boundaries of the LZ1, LZ2 and LZ3 zones plotted.  This would make it easier to interpret the results. In most of the legends ( figures and diagrams) there is lack of description of units e.g. currency or area units.
The author should also check the quality of the translation, which in my opinion is not always correct.
The described remarks do not drastically influence the quality of the work, which after minor corrections should be published.

Author Response

Dear reviewer-

Thank you for taking your time to read my manuscript and forward your invaluable inputs, I have put the responses to your comments. The explained responses and modifications made in the manuscript such as paragraphs, figures, and tables are also copied and pasted in this response template. Moreover, for you to easily track changes, the line numbers are put at the end of each response (whenever necessary). Line numbers are based on the improved version. Also, for easy pass from one question to another, your questions are highlighted with yellow.

With kind regards.

Reviewer 2 Report

The paper is interesting. Its structure is proper, but there are a lot of editing mistakes. The numbering of different sections should be improved. There are also some unnecessary spaces in the text.

The paper contains a survey during 2013-2017. We have 2022 now (5 years later). The analysis is done on archival data and does not relate to the actual situation on the market. That is why, in my opinion, the data should be updated, and the paper should be reconsidered afterward.

Author Response

(The authors gave the same response as above.)

Reviewer 3 Report

An interesting article, which is based on the general basis of this issue, an article of a rather descriptive nature. An article that relates to the described locality, without broader connections and contexts for other territories, without further comparison. The article is interesting, but it is not a presentation of fundamental scientific results.

  • Abstract: the abstract does not have a good structure, the text lacks context "problems - goal - methods of scientific work - basic results.
    In the abstract, there are dots next to the numbers.
  • Wrong chapter numbering.
  • It is advisable to write the literature according to the recommended template.
  • The article lacks a broader context of the issue, comparison or comparison with similar cities, the potential for further research - what needs to be further focused (for example, in the Discussion chapter).

Author Response

(The authors gave the same response as above.)

Reviewer 4 Report

The article is a well-developed source of knowledge on changes in the structure of land use in urban areas. The authors once again proved the known truth that there is a close relationship (feedback) between the intensity of land use and its market value. In this respect, it is not a new "discovery". However, diligently conducted empirical research and analysis of the obtained results deserve attention.

 The issue of the initial value of land offered for sale and built-up real estate offered for rent requires a more detailed explanation. How does the commune as the owner determine their offer prices? Do municipal authorities use the services of professional property appraisers? If so, are there any real estate appraisal rules / standards?

Is the property tax based on the value of the property?

Is there a system of help for people in a difficult economic situation, e.g. by regulating / differentiating the rental rates?

Introducing these issues to the article will increase its cognitive value in the context of the specificity of implementing the principles of sustainable spatial management in selected areas (countries).

Author Response

(The authors gave the same response as above.)

Round 2

Reviewer 3 Report

The correction is well done.

Author Response

Dear reviewer-

Thank you again for going through the manuscript and your confirmation that your comments were addressed. Moreover, we are submitting the manuscript to a professional language editor under the MDPI

Kind regards,

Amanuel